# Tribological Properties of Micro-Groove Cemented Carbide by Laser Processing

**DOI:** 10.3390/mi12050486

**Published:** 2021-04-25

**Authors:** Guoqiang Zheng, Youxi Lin

**Affiliations:** School of Mechanical Engineering and Automation, Fuzhou University, Wulongjiangbei Avenue Road 2 of Fuzhou University City, Fuzhou 350108, China; zgqfzu@163.com

**Keywords:** anti-friction mechanism, micro-groove, friction coefficient, ability to remove metal debris

## Abstract

Tool wear is the main factor of tool failure in cutting difficult-to-machine materials. This paper aims to analyze the anti-friction mechanism of laser machining micro-groove cemented carbide. Firstly, micro-grooves were prepared on the cemented carbide surface by laser processing. Secondly, we conducted an analysis of the mechanical properties of laser texturing by measuring hardness. Finally, we studied the anti-friction mechanism of micro-grooves by a wear test (ASTM G133-05). Results show that surface hardness increases after laser treatment. The friction coefficient and surface wear of micro-groove cemented carbide are significantly reduced compared with the conventional surface. The friction coefficient of PE and OB decreased by 20.6% and 10.7%, respectively. It is found that the direction of micro-grooves determines whether metal debris can be removed—the stronger the ability to remove metal debris, the better the tribological properties of the micro-groove surface.

## 1. Introduction

Cemented carbide, an alloy made of tungsten carbide and cobalt by powder metallurgy, is widely used as a tool material for turning tools, milling cutters, and drill bits because of its high hardness, good strength, wear resistance, and corrosion resistance [1,2,3,4]. However, severe friction at the tool–chip interface causes a high contact pressure and a large amount of cutting heat when cutting difficult-to-machine materials such as titanium alloy, stainless steel, and high-manganese steel, which reduces the service life and machining efficiency of cemented carbide tools [5,6,7]. Coatings and cutting fluids are the conventional methods to reduce tool wear [8,9,10]. However, with the deepening of research, it becomes more challenging to develop new coating materials. Simultaneously, introducing cutting fluids will bring up economic, environmental, health, and safety concerns.

In recent years, many studies have shown that micro-texture is an effective method for reducing the friction of the contact surface. Micro-texture can accommodate wear debris [11,12], store lubricating oil [13,14], and enhance the bearing capacity of the lubricating interface [15,16,17] to enhance the tribological properties of the material surface. Laser processing is widely used in the fabrication of micro-texture due to its high efficiency, high precision, and zero pollution [18,19,20]. At the same time, laser processing is a heat treatment process, and different laser processing parameters (power, scanning speed, frequency) will cause different laser ablation situations [21]. It will inevitably change the material properties, such as roughness [22] and mechanical properties [23], thereby affecting the tribological properties of the material surface.

Many researchers have observed a significant improvement in cutting performance by applying micro-texture to tool surfaces. Liu et al. investigated the effects of micro-texture on the cutting performance of ceramic tools. The results showed that the micro-texture tool has a smaller friction coefficient, better wear resistance, and more uniform stress distribution [24]. Xie et al. prepared micro-grooves on the rake face of CBN tools by micro-grinding and studied the effect of micro-grooves on cutting performance. It was pointed out that the micro-groove CBN tool has a better heat dissipation performance, which can effectively reduce the cutting temperature and prolong the tool life [25]. Su et al. used micro-texture PCD tools to turn titanium alloy bar. It was found that the friction force and chip adhesion area of the micro-texture PCD tool are lower than conventional tools, and the maximum width of chip adhesion is reduced by 23.6% [26]. Li et al. analyzed the influence of micro-texture on the cutting performance of PCB tools. Cutting experiments and finite element analysis results showed that reasonable micro-texture could reduce the cutting force, reduce tool wear, and improve the machining quality [27]. Sawant et al. studied the cutting performance of a micro-texture tool and pointed out that micro-texture can reduce the curvature radius of the chip and avoid the formation of banded chips, which decreases the contact length between the chip and the rake face, thereby reducing the cutting temperature [28]. Kummel et al. revealed that the appropriate micro-texture tool could increase the stability of BUE and avoid its periodic adhesion and peeling, which not only reduces the tool wear but also improves the machining quality [29].

In conclusion, prior researchers mainly focused on cutting performance. The innovation of this paper is to analyze the anti-friction mechanism. To solve the problem, firstly, micro-grooves were prepared on a cemented carbide surface by laser processing. Secondly, we conducted an analysis of the mechanical properties of laser texturing by measuring hardness. Finally, we studied the anti-friction mechanism of micro-grooves by a wear test. It is found that the direction of micro-grooves determines whether metal debris can be removed—the stronger the ability to remove metal debris, the better the tribological properties of the micro-groove surface.

## 2. Materials and Methods

### 2.1. Fabrication of Micro-Groove Cemented Carbide

YG8 cemented carbide was selected as the test material in this study. Table 1 shows the physical and mechanical properties of cemented carbide. The dimensions of cemented carbide were 5 mm × 15 mm × 25 mm. A micro-groove texture was fabricated on the cemented carbide surface by a fiber laser (YLP-CX-20). The fiber laser processing parameters were as follows: laser power 16KW, scanning speed 100 mm/s, laser frequency 20 Hz, scanning times 1 time.

Five types of micro-groove were generated on the cemented carbide surface: perpendicular micro-groove cemented carbide (PE), oblique micro-groove cemented carbide (OB), parallel-1 micro-groove cemented carbide (PA-1), parallel-2 micro-groove cemented carbide (PA-2), and parallel-3 micro-groove cemented carbide (PA-3). The SEM images of the surface morphology of micro-groove cemented carbide are presented in Figure 1. Table 2 provides the geometric parameters of micro-grooves, and the meaning of each geometric parameter is shown in Figure 1c.

After laser processing, metallographic sandpaper (1200 #) was used to polish the micro-groove cemented carbide sample carefully, and ultrasonic cleaning was carried out in dehydrated ethanol to remove the influence of surface impurities on the experimental results.

### 2.2. Hardness Test

The hardness of micro-groove cemented carbide was measured by an HR-150A Rockwell hardness tester equipped with a diamond pressure head under loading of 600 N and a holding time of 10 s. To ensure the accuracy of the measurement data, six points were selected on each sample surface to measure the hardness, and the position of the measurement point remained unchanged. The first measurement data were not included, and the average value of the last five measurements was taken as the hardness of the sample.

### 2.3. Wear Test

The wear test (ASTM G133-05) was carried out on a universal mechanical tester (UMT-II), and the experimental setup is shown in Figure 2. The testing machine is a high-end material testing instrument that integrates rotating, reciprocating, ring block, and other measurement methods. It can test the friction coefficient and wear of different types of materials. Wear tests took reciprocating friction: the upper friction pair was a 400C stainless steel ball, diameter 12.7 mm, Rockwell hardness value 58HRC; the lower friction pair was the prepared micro-groove cemented carbide. The upper friction pair was still and applied a 25 N load to the lower friction pair during the test. The lower friction pair moved in a reciprocating motion along the Y direction with the workbench, realizing the dry sliding friction between the stainless steel ball and micro-groove cemented carbide. The wear test parameters are shown in Table 3.

## 3. Results and Discussion

### 3.1. Mechanical Properties of Laser Processing Micro-Groove Cemented Carbide

Figure 3 presents the hardness measurement results of conventional surface cemented carbide CO, the laser-treated surface LA, and micro-groove cemented carbide with different geometric parameters. Figure 3 shows that LA’s surface hardness is higher than that of CO, so laser processing’s thermal effect can increase the hardness of the material, which affects the mechanical properties of the material. The hardness of micro-groove cemented carbide is smaller than that of conventional surface cemented carbide, and the OB micro-groove cemented carbide is the smallest.

Due to the self-quenching of the material surface during laser processing, the hardness of the laser-treated surface increases, which will improve the wear resistance of the material. More specifically, the surface temperature rises rapidly above the phase transition point when the high-energy laser beam passes through the material surface. Surface cooling occurs rapidly due to the excellent heat transfer performance of the metal matrix after laser beam removal. It can also be seen that the micro-grooves will weaken the cemented carbide strength to a certain extent, so the hardness of micro-groove cemented carbide is small.

### 3.2. Friction Coefficient of Micro-Groove Cemented Carbide

To compare the tendency of the friction coefficient for different micro-grooves more clearly, the average value of the friction coefficient in each minute is taken as the friction coefficient of the sample in that minute. Figure 4 shows friction coefficients as functions of the time with conventional surface and micro-groove cemented carbides. These results show that conventional surface cemented carbides and micro-groove cemented carbides experienced the running-in stage and stable wear stage. The friction coefficient increases first and then tends to be stable; their running-in times are different. Micro-groove cemented carbide has a shorter running-in time; the PE running-in time is three minutes; other micro-groove cemented carbides’ running-in time is about eight minutes; the CO running-in time is eleven minutes. It can be seen that the micro-grooves can shorten the running-in time to enter the stable wear stage of the low wear rate in advance, reducing the wear amount.

These results also show that conventional surface cemented carbide had a smaller friction coefficient in the first four minutes of the experiment. This is mainly because the micro-grooves increase the surface roughness of the material, which deteriorates the surface tribological properties. However, with the continuous sliding pressure of the stainless steel balls, the surface roughness of micro-groove cemented carbide gradually decreases. Therefore, the friction coefficients of PE and OB are always smaller than conventional surface cemented carbide from the sixth minute.

It can be seen from Figure 4 that each sample reached the stable wear stage 10 min later. The stable wear stage’s average friction coefficient was calculated to reflect the effect of micro-grooves on the friction coefficient more clearly. The average friction coefficient in 10–15 min is shown in Figure 5. It can be seen that during this period, the friction coefficient of conventional surface cemented carbide is 0.533, the friction coefficient of PE is 0.423, the friction coefficient of OB is 0.476, the friction coefficient of PA-1 is 0.526, the friction coefficient of PA-2 is 0.548, and the friction coefficient of PA-3 is 0.524. The results show that PE’s and OB’s friction coefficient decreases by 20.6% and 10.7%, respectively, and there is no significant decrease in parallel micro-grooves. It can also be seen that the friction coefficient increases as the micro-groove spacing decreases.

### 3.3. Wear Morphology of Micro-Groove Cemented Carbide

Figure 6 shows the SEM topography of different kinds of cemented carbide. The names of the developed cemented carbides with micro-grooves in Figure 6a–f correspond to CO, PE, OB, PA-1, PA-2, and PA-3, respectively (Table 2).

It can be found that there is a large area of metal adhesion and a severe scratch on the surface of conventional surface cemented carbide; the micro-groove cemented carbide surface metal adhesion is small and there is no obvious scratch. However, it can be seen from the large amount of metal debris in the micro-grooves that there is abrasive wear in the friction process, but the degree of abrasive wear is light. From the results of the experiments described above, we know that the wear mechanism of conventional surface and micro-groove surface is typical adhesive wear and abrasive wear, and micro-grooves can reduce the wear of cemented carbide.

Based on the experimental results in Figure 6, when the micro-groove direction is consistent with the friction direction (PE), there are no evident adhesion and scratches on the surface of cemented carbide, and metal debris in the micro-grooves is presented as powdered. When the angle between the micro-groove direction and friction direction is 45° (OB), a small amount of adhesion appears on the surface of cemented carbide, and metal debris in the micro-grooves is also presented as powdered. When the micro-groove direction is perpendicular to the friction direction (PA-1, PA-2, and PA-3), a large adhesion area appears on the surface of cemented carbide, and metal debris in the micro-grooves is presented as blocky. From the above observations, it is known that the micro-groove direction affects the wear morphology of cemented carbide. When the micro-groove direction is consistent with the friction direction, the surface wear is lighter.

Figure 6d–f show that the wear morphology of the cemented carbide surface with different micro-groove spacings under the micro-groove direction is perpendicular to the friction direction. It can be observed that micro-grooves’ metal debris increases with the micro-groove spacing ranging from 30 μm to 70 μm, which means that the degree of abrasive wear increases. It can be deduced that the micro-groove spacing affects the surface wear of cemented carbide—the smaller the spacing, the lower the surface tribological properties.

### 3.4. Anti-Friction Mechanism of Micro-Groove Cemented Carbide

Figure 7a,b show the schematic diagram of PA and PE micro-groove cemented carbides’ anti-friction mechanism. To explain the anti-friction mechanism more clearly, Figure 7a is the A-A cross-section of the 3D schematic of the PE wear test (see Figure 8). It can be seen from Figure 7 that there are small amounts of metal debris in the PA and PE micro-grooves at the initial stage of the wear test. As the wear test is carried out, new metal debris is constantly produced. Excess metal debris in the PA micro-grooves will move with the stainless steel ball and accumulate at the friction stroke’s start and end. There is only a tiny amount of metal debris in the micro-grooves within the stroke; however, excess metal debris in the PE micro-grooves cannot be removed in time, and the micro-grooves in the friction stroke are filled with metal debris.

It is known from the above results that micro-grooves can accommodate metal debris between the friction pairs to avoid metal debris damaging the material surface. The tribological properties of micro-groove cemented carbide depend on whether it has metal debris removal ability. When the micro-groove direction is consistent with the friction direction (PE), metal debris can be removed in time with the friction pair’s movement to ensure that the micro-grooves can accommodate metal debris throughout the process, and the corresponding wear morphology is shown in Figure 6b. When the micro-groove direction is perpendicular to the friction direction (PA), the micro-grooves have no metal debris removal ability. When the micro-grooves are filled with metal fragments, the micro-groove morphology disappears, which leads to the inability to accommodate new metal debris, as shown in Figure 6d–f. Excess metal debris can only exist between the friction pairs, resulting in adhesive wear and abrasive wear on the material surface.

The above phenomenon can also be explained from the perspective of the friction coefficient. As shown in Figure 5, PE had the best metal debris removal ability, and the friction coefficient decreased by 20.6%. OB has a certain metal debris removal ability, and the friction coefficient is reduced by 10.7%; PA-1, PA-2, and PA-3 have no metal debris removal ability, so the friction coefficient is similar to that of conventional surface cemented carbide.

## 4. Conclusions

(1)Rapid heating and cooling of the metal surface can increase the hardness of materials during laser processing, which will improve the wear resistance and tribological properties of the materials. It can also be seen that micro-grooves will weaken the cemented carbide strength to a certain extent.(2)Micro-grooves can shorten the running-in time, which PE micro-grooves and other micro-grooves shortened by 8 min and 3 min, respectively. Shortening the running-in time can assist in entering the stable wear stage of the low wear rate in advance, reducing the wear amount.(3)The direction and spacing of micro-grooves will affect the friction coefficient. The friction coefficient of PE and OB decreased by 20.6% and 10.7%, respectively. There was no significant decrease in the friction coefficient of parallel micro-grooves, and the friction coefficient increases as the micro-groove spacing decreases.(4)Micro-grooves can accommodate metal debris between friction pairs, avoiding damaging the material surface, to enhance the tribological properties of the material surface. The direction of micro-grooves determines whether metal debris can be removed—the stronger the ability to remove metal debris, the better the tribological properties of the micro-groove surface.

## Figures and Tables

**Figure 1 micromachines-12-00486-f001:**
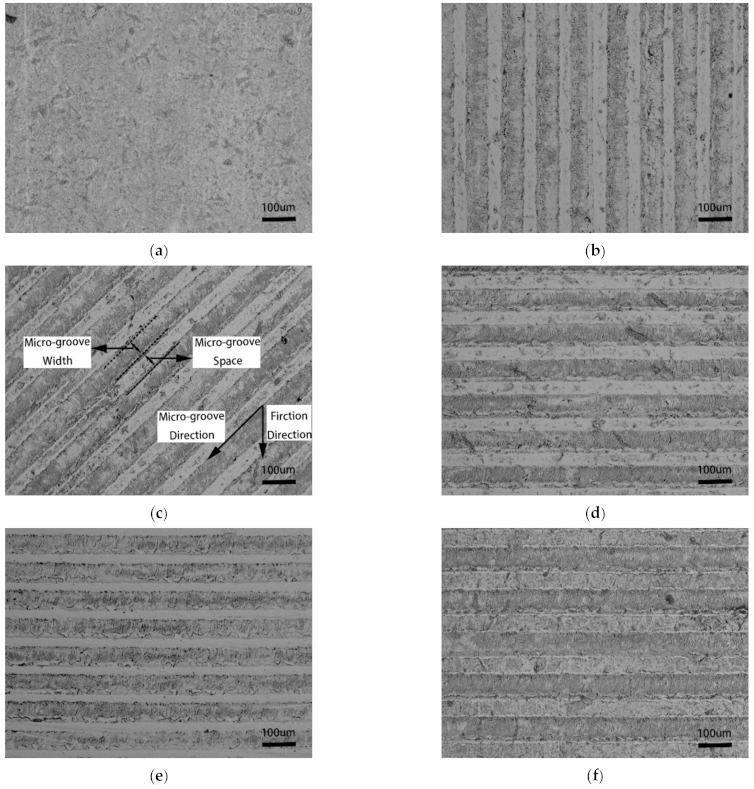
The surface morphology of micro-groove cemented carbide. (**a**) Conventional surface; (**b**) PE micro-grooves; (**c**) OB micro-grooves; (**d**) PA-1 micro-grooves; (**e**) PA-2 micro-grooves; (**f**) PA-3 micro-grooves.

**Figure 2 micromachines-12-00486-f002:**
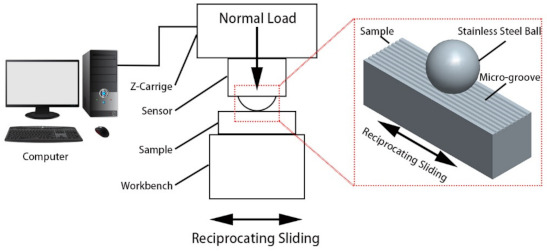
Experimental setup for wear test (UMT-II).

**Figure 3 micromachines-12-00486-f003:**
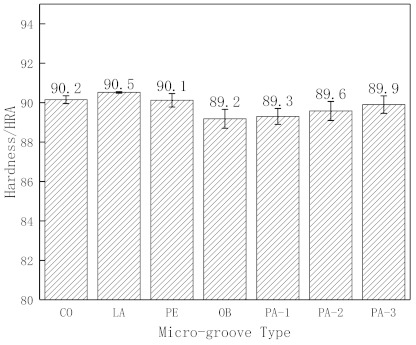
Hardness measurement results.

**Figure 4 micromachines-12-00486-f004:**
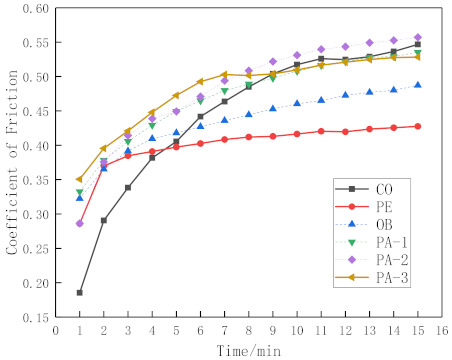
Comparison of friction coefficient.

**Figure 5 micromachines-12-00486-f005:**
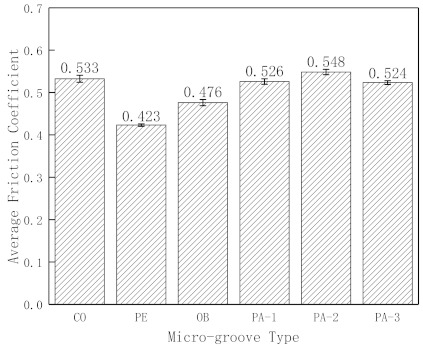
Average values of friction coefficient.

**Figure 6 micromachines-12-00486-f006:**
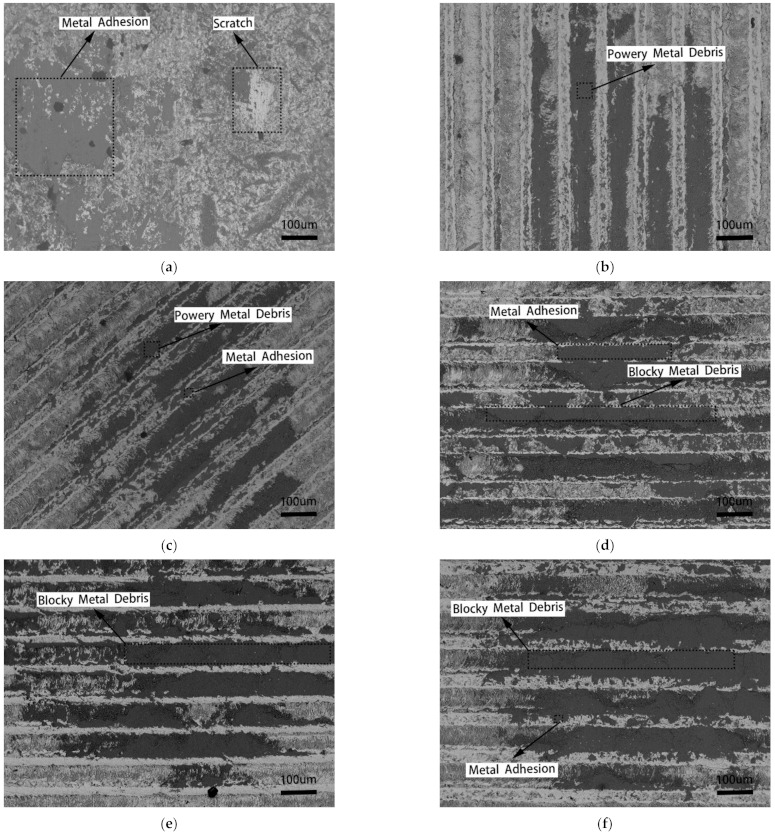
The wear morphology of micro-groove cemented carbide. (**a**) Conventional surface; (**b**) PE micro-grooves; (**c**) OB micro-grooves; (**d**) PA-1 micro-grooves; (**e**) PA-2 micro-grooves; (**f**) PA-3 micro-grooves.

**Figure 7 micromachines-12-00486-f007:**
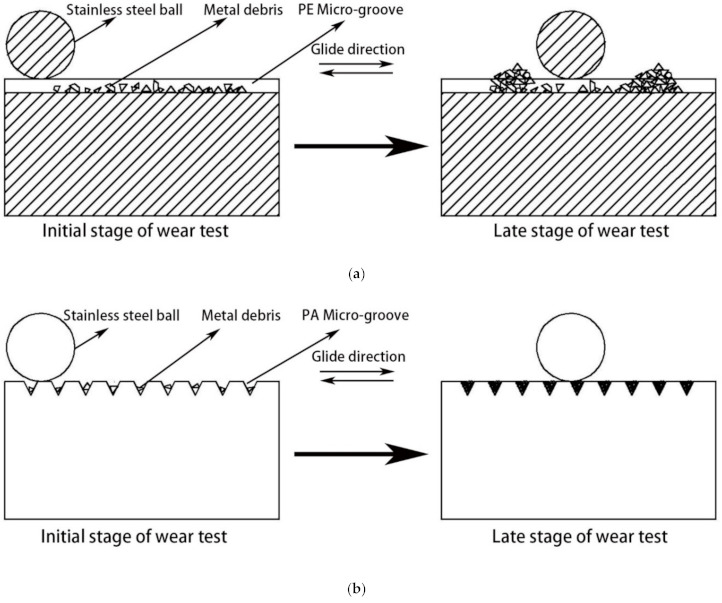
The anti-friction mechanism of micro-groove cemented carbide. (**a**) A-A cross-section of PE micro-grooves; (**b**) PA micro-grooves.

**Figure 8 micromachines-12-00486-f008:**
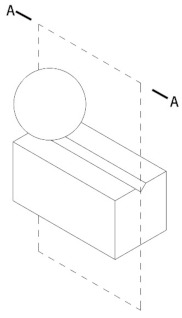
The 3D schematic of the PE wear test.

**Table 1 micromachines-12-00486-t001:** The physical and mechanical properties of cemented carbide.

Density (g/cm^3^)	Hardness (HRA)	Impact Ductility (J/cm^2^)	Bending Strength (N/mm^2^)
14.74	89	2.5	2710

**Table 2 micromachines-12-00486-t002:** The geometric parameters of micro-grooves.

Type	Micro-Groove Width (μm)	Angle between Micro-Groove and Friction Direction (°)	Micro-Groove Space (μm)
PE	50	0	50
OB		45	50
PA-1		90	50
PA-2		90	30
PA-3		90	70

**Table 3 micromachines-12-00486-t003:** The wear test parameters.

Load (N)	Sliding Speed (mm/s)	Time (min)	Reciprocating Distance (mm)
25	18	15	4

## Data Availability

The data presented in this study are available in article.

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
