# Peer review of "Tribological Properties of Micro-Groove Cemented Carbide by Laser Processing"

_micromachines, 2021, doi:10.3390/mi12050486_

Round 1
Reviewer 1 Report
- Error bar should be added for figure 3 and 5.
- There should be description on how friction coefficient is calculated. Raw measurement data needs to be shown as well.
- In Figure 5, the name of PA and PE should be switched. It should be PA-1, PA-2, and PA-3.
- Figure 8 is introduced before figure 7. The numbering should be adjusted.
- Most of conclusions are based on figure 6 with observation only. It is difficult to judge as the differences are not obvious. There should be labeling on figures matching the description that could help understanding. More importantly, there should be measurements on the wear so that quantitative comparison can be made. In addition, the SEM images matching figure 8 should be provided so that the described anti-friction mechanism can be confirmed in reality.
Author Response
Dear reviewer,
On behalf of my co-authors, we thank you very much for giving us an opportunity to revise our manuscript. We appreciate very much your positive and constructive comments and suggestions on our manuscript entitled Tribological Properties of Micro-groove Cemented Carbide by Laser Processing (micromachines-1194931). To address the critiques, we revised our manuscript according to comments. The attached file contains the corresponding revisions. We would like to express our great appreciation to you for your comments on our paper.
Looking forward to hearing from you. Thank you and best regards.
Yours sincerely,

Reviewer 2 Report
Paper need to included all the changes and suggestions given in the attached file

Author Response

(The authors gave the same response as above.)

Round 2
Reviewer 1 Report
Comments are addressed properly.
Reviewer 2 Report
Paper is OK